Lack of riluzole efficacy in the progression of the neurodegenerative phenotype in a new conditional mouse model of striatal degeneration

Kreiner Grzegorz kreiner@if-pan.krakow.pl
Rafa-Zabłocka Katarzyna
Chmielarz Piotr
Bagińska Monika
Nalepa Irena
Institute of Pharmacology, Polish Academy of Sciences, Dept. Brain Biochemistry , Kraków , Poland
Silva Pedro
Electronic publication date: 2017 Apr 27
Publication date: 2017
Volume: 5
Electronic Location ID: e3240
Received 2016 Oct 28; Accepted 2017 Mar 28
Copyright: ©2017 Kreiner et al.
Copyright year: 2017
Copyright holder: Kreiner et al.
License: This is an open access article distributed under the terms of the Creative Commons Attribution License, which permits unrestricted use, distribution, reproduction and adaptation in any medium and for any purpose provided that it is properly attributed. For attribution, the original author(s), title, publication source (PeerJ) and either DOI or URL of the article must be cited.
License URL: https://creativecommons.org/licenses/by/4.0/

Keywords: Riluzole, Neurodegeneration, Huntington’s disease, Transgenic models, Neurogenesis, Neuroprotection

Funding: National Science Centre Opus2, 2011/03/B/NZ7/05949 This work was supported by the National Science Centre (grant Opus2, 2011/03/B/NZ7/05949). The funders had no role in study design, data collection and analysis, decision to publish, or preparation of the manuscript.

==============================
Background

Huntington’s disease (HD) is a rare familial autosomal dominant neurodegenerative disorder characterized by progressive degeneration of medium spiny neurons (MSNs) located in the striatum. Currently available treatments of HD are only limited to alleviating symptoms; therefore, high expectations for an effective therapy are associated with potential replacement of lost neurons through stimulation of postnatal neurogenesis. One of the drugs of potential interest for the treatment of HD is riluzole, which may act as a positive modulator of adult neurogenesis, promoting replacement of damaged MSNs. The aim of this study was to evaluate the effects of chronic riluzole treatment on a novel HD-like transgenic mouse model, based on the genetic ablation of the transcription factor TIF-IA. This model is characterized by selective and progressive degeneration of MSNs.

Methods

Selective ablation of TIF-IA in MSNs (TIF-IAD1RCre mice) was achieved by Cre-based recombination driven by the dopamine 1 receptor (D1R) promoter in the C57Bl/6N mouse strain. Riluzole was administered for 14 consecutive days (5 mg/kg, i.p.; 1× daily) starting at six weeks of age. Behavioral analysis included a motor coordination test performed on 13-week-old animals on an accelerated rotarod (4–40 r.p.m.; 5 min). To visualize the potential effects of riluzole treatment, the striata of the animals were stained by immunohistochemistry (IHC) and/or immunofluorescence (IF) with Ki67 (marker of proliferating cells), neuronal markers (NeuN, MAP2, DCX), and markers associated with neurodegeneration (GFAP, 8OHdG, FluoroJade C). Additionally, the morphology of dendritic spines of neurons was assessed by a commercially available FD Rapid Golgi Stain™ Kit.

Results

A comparative analysis of IHC staining patterns with chosen markers for the neurodegeneration process in MSNs did not show an effect of riluzole on delaying the progression of MSN cell death despite an observed enhancement of cell proliferation as visualized by the Ki67 marker. A lack of a riluzole effect was also reflected by the behavioral phenotype associated with MSN degeneration. Moreover, the analysis of dendritic spine morphology did not show differences between mutant and control animals.

Discussion

Despite the observed increase in newborn cells in the subventricular zone (SVZ) after riluzole administration, our study did not show any differences between riluzole-treated and non-treated mutants, revealing a similar extent of the neurodegenerative phenotype evaluated in 13-week-old TIF-IAD1RCre animals. This could be due to either the treatment paradigm (relatively low dose of riluzole used for this study) or the possibility that the effects were simply too weak to have any functional meaning. Nevertheless, this study is in line with others that question the effectiveness of riluzole in animal models and raise concerns about the utility of this drug due to its rather modest clinical efficacy.

Introduction

Huntington’s disease (HD) is a rare (1:10000) familial autosomal dominant neurodegenerative disorder caused by an expanded stretch of polyglutamine (polyQ) repeats in the protein huntingtin (Hannan, 2005) and characterized by progressive degeneration of medium spiny neurons (MSNs) located in the striatum. The disease inevitably culminates with death and cures to at least retard its progression are unavailable so far. Currently available treatments are limited to alleviating some of the symptoms, mainly involuntary movements, associated with the disease. Despite the known origin, there is a lack of understanding of the complex pathogenesis of HD, which affects multiple functions and regulatory pathways, making the development of efficient therapeutics challenging (Kazantsev & Hersch, 2007). Classic pharmacological models of HD are based on applying a neurotoxin, 3-nitropropionic acid (3-NP) (Tunez et al., 2010); however, this approach leads to immediate neuronal death, which substantially narrows the opportunity to observe the pathological changes associated with the slow neurodegenerative process. On the other hand, many transgenic animal models of HD, even though created by replicating the same genetic malfunction directly responsible for HD in humans, do not fully recapitulate the HD-like phenotype, including profound neuronal loss (or at least not to the expected extent) (Kreiner, 2015).

Designed cell therapies for neurodegenerative diseases are mostly based on the replacement of lost neurons through transplantation or activation of neuronal progenitor cells (Emsley et al., 2005). In rodent models of HD, induced neurogenesis in MSNs is thought to be evoked primarily due to neuronal precursors derived from the subventricular zone (SVZ) of the lateral ventricles. The SVZ represents the largest reservoir of adult stem-like progenitors and in normal conditions gives rise to new olfactory bulb interneurons (Bonfanti & Peretto, 2007). Stimulation of postnatal neurogenesis is being considered as a potential therapeutic target in several neurodegenerative diseases including HD (Abdipranoto et al., 2008; Lindvall & Kokaia, 2010; Ransome, Renoir & Hannan, 2012). One of the drugs of potential interest for the treatment of HD is riluzole, already approved for the treatment of amyotrophic lateral sclerosis (ALS) (Miller, Mitchell & Moore, 2012). Riluzole, by interfering with glutamatergic neurotransmission, reduces excitotoxicity and acts as a positive modulator of adult neurogenesis, promoting replacement of damagedMSNs; however, whether it has any clinical meaning remains not clear (Katoh-Semba et al., 2002; Squitieri et al., 2008; Veyrac et al., 2009). It was also shown that riluzole treatment can result in enhancement of damaged neurite formation potentially leading to functional recovery of motoneurons in rat model of L4-6 root avulsion (Bergerot et al., 2004). Based on experimental data coming from cell and animal research, the classic pharmacological mechanism of its action is related to so-called excitotoxic hypothesis of neurodegeneration. Namely, riluzole can inhibit the release of glutamic acid most likely due to the inactivation of voltage-dependent sodium channels on glutamatergic nerve terminals, as well as activation of a G-protein-dependent signaling pathways (Doble, 1996). Another postulated mechanism associated with beneficial role of riluzole application is related to observed increase of serum concentrations of brain-derived neurotrophic factor (BDNF) (Katoh-Semba et al., 2002), which neurotrophic factor is known to be significantly diminished in the brains of HD patients, and its level seems to be correlated with diseases onset progression and severity (Gauthier et al., 2004). Moreover, riluzole was shown to be effective in attenuating several clinically relevant symptoms in a variation of an animal MPTP model representing the early phase of Parkinson’s disease (PD) (Verhave et al., 2012). Nevertheless, there are still concerns about its utility due to rather modest clinical efficacy (Miller, Mitchell & Moore, 2012).

To address this question, we applied a novel approach using a mouse model of HD-like phenotype, based on the activation of an endogenous suicide mechanism achieved by genetic ablation of the transcription factor TIF-IA, an essential regulator of polymerase I activity (Kreiner et al., 2013). Inactivation of TIF-IA blocks the synthesis of ribosomal RNA, leading to nucleolar disruption and p53-mediated apoptosis (Yuan et al., 2005). Loss of TIF-IA in neuronal progenitor cells results in mice born without a brain (Parlato et al., 2008), but when it is lost in mature neurons, the major features of the neurodegenerative process are recapitulated. Namely, inactivation of the TIF-IA gene in striatal MSNs (TIF-IAD1RCre mice) recapitulates the phenotypic alterations associated with selective striatal neurodegeneration (occurring in 13-week-old mice), including increased oxidative damage and inflammatory response, finally leading to MSN cell death and resulting in an HD-like phenotype (Kreiner et al., 2013). In particular, we have shown that 13-week-old TIF-IA D1RCre mice were characterized by profoundly enhanced expression of astro- and microglia markers (GFAP, CD11b), several oxidative stress markers (8-hydroxydeoxyguanosine, 8-OHdG; nitrosylated tyrosine, NITT, neuroketals, NK) as well as TUNEL+ cells. The MSNs cells were progressively lost over the time as visualized by NeuN and D1R immunohistochemical stainings. These cellular events were associated with motor impairment assessed by rotarod and clasping behavior (Kreiner et al., 2013). In contrast to the majority of other models of neurodegeneration, TIF-IAD1Cre-mutant mice are characterized by the progressive degeneration of targeted neurons over a long period of time (several weeks), mimicking the typical hallmark of the disease (Kreiner et al., 2013).

Materials & Methods

The summary of experimental design is illustrated on the chart (Fig. 1).

Figure 1 The summary of experimental design.

The mice were treated with riluzole (5 mg/kg, i.p.) starting at the 6th week of age. The phenotype of riluzole treated vs. non-treated animals was compared at the 13th week of age on behavioral and immunohistochemical level, when the effects of the mutation are clearly manifested in neuronal cell loss and behavioral impairment. Ki67 expression and dendritic spines morphology were assessed at the 9th week when cell loss in mutant animals have not yet been observed.

Mice

Selective ablation of TIF-IA in the MSNs (TIF-IAD1RCre mice) was achieved by Cre/loxP recombination in the C57Bl/6N mouse strain. Transgenic mice hosting Cre recombinase under the dopamine 1 receptor (D1R) promoter were crossed with animals harboring the floxed TIF-IA gene as described previously (Kreiner et al., 2013). Mutant TIF-IAD1RCre mice were kept together with their control (Cre-negative) littermates in self-ventilated cages (Allentown, Inc., Allentown, NJ, USA) under standard laboratory conditions (12 h light/dark cycle, food and water ad libitum). This study was carried out in strict accordance with the recommendations in the Guide for the Care and Use of Laboratory Animals of the National Institutes of Health. The protocol for the behavioral study was approved by the Animal Ethical Committee at the Institute of Pharmacology, Polish Academy of Sciences (Permit Number: 951, issued: June 28, 2012).

Drug treatment

After genotyping, the 6-week-old mice were divided into four experimental groups: control+VEH (con/VEH), control+RIL (con/RIL), mutant+VEH (mut/VEH), mutant+RIL (mut/RIL), receiving either riluzole (RIL; 5 mg/kg, i.p.; Sigma-Aldrich Chemical Co., St. Louis, USA) or vehicle (VEH; 10% DMSO) for 14 consecutive days (1× daily). These doses did not influence daily cage normal behavior observed in riluzole and vehicle treated mice.

Behavioral analysis

A coordination test was performed on 13-week-old animals on an accelerated rotarod (Ugo Basile, Italy). The assessment was preceded by training session, one day before the experiment (5 min on the rotating rod, constant speed). During the experiment the time spent on the accelerating rod (4–40 r.p.m. within 5 min) was measured. Additionally, the weight of the animals was consistently monitored during the time of drug application and on the day before the behavioral test.

Immunohistochemistry

To visualize the potential effects of drug treatment, the striata of animals were subject to post-mortem staining using immunohistochemistry (IHC) and/or immunofluorescence (IF) with specific markers as described previously (Chmielarz et al., 2013; Kreiner et al., 2013). Briefly, the mice were sacrificed by cervical dislocation, and their brains were excised, fixed overnight in 4% paraformaldehyde (PFA), dehydrated, embedded in paraffin and sectioned on a rotary microtome on 7 µm thick slices. Chosen sections from corresponding regions of the striatum in mutant and control animals were incubated overnight at 4 °C with primary anti-NeuN (1:500, cat. no MAB377; Millipore, Billerica, MA, USA), anti-MAP2 (1:1000, Abcam; cat. no ab5392), anti-doublecortin (DCX) (1:100, Abcam; cat. no ab135349), anti-GFAP (1:500, cat. no AB5541; Millipore, Billerica, MA, USA), and anti-8OHdG (1:200, cat. no AB5830; Millipore, Billerica, MA, USA) antibodies. Visualization of antigen-bound primary antibodies was carried out using a proper biotinylated secondary antibody together with the Avidin–Biotin Complex (ABC; Vector Laboratories, Burlingame, CA, USA) followed by diaminobenzidine treatment (DAB; Sigma-Aldrich, St. Louis, MO, USA) or an anti-rabbit Alexa-488 or Alexa-594-coupled secondary antibody (Invitrogen, Carlsbad, CA, USA). FluoroJade C (cat. no AG325; Millipore, Billerica, MA, USA) staining was performed according to manufacturer’s protocol. Briefly, after deparaffinization and initial incubation in 0.06% KMNO4 (10 min.) the slides were rinsed in distilled water and immersed for 10 min. in 0.001% solution of FluoroJade C dissolved in 0.1% acetic acid vehicle.

Quantification of Ki67 expression was done by counting all Ki67-positive cells on adjacent sections from n = 4–6 animals of each genotype/treatment in a single-blind experiment (an investigator did not know which samples belong to which genotype/treatment).

Dendritic spine morphology

Morphological analysis of dendritic spines was assessed as described previously (Chmielarz et al., 2015). Briefly, following extraction, the brains were rinsed in distilled water, impregnated with the use of the FD Rapid Golgi Stain™ Kit (FD NeuroTechnologies, Columbia, MD, USA), and incubated in 30% sucrose for 3–7 days. Vibratome (Leica, Wetzlar, Germany) sections were cut to 100 µm thick and mounted on Super Frost Plus slides (Thermo Scientific, USA) and stained using solutions provided in the kit. The dendritic spines were counted on the dorsal striatum between Bregma 1.1 and 0.0. Dendritic spines were counted on at least 10 µM long fragments of 3rd and 4th row dendrites. There were three pieces counted from each neuron and five neurons counted for each animal. Only completely stained neurons not obscured by neighboring neurons within the striatum were considered. Spine counting and optical imaging were performed by an experimenter blind to the genotype of the animal on a Nikon Eclipse 50i (Nikon, Tokyo, Japan) equipped with a CCD camera connected to a computer equipped with NIS Elements BR 30 software.

Statistical analysis

Statistical analysis was performed with Graph Pad Prism 5.01. Data were evaluated by 2-way analysis of variance (2-way ANOVA) followed by Bonferroni test for comparison of biologically relevant groups.

Results

Enhancement of cell proliferation observed in TIF-IAD1Cre mutant mice and after riluzole treatment

The expression of Ki67 showed substantial enhancement in the region of the SVZ in non-treated 9-week-old TIF-IAD1Cre mice and all riluzole-treated animals (Figs. 2A–2B). Double immunofluorescent staining revealed that the number of cells labelled with Ki67 (marker of cell proliferation) in SVZ co-localize with the MAP2 (microtubule-associated protein-2, neuronal marker) (Fig. 2C) or doblecortin (DCX) positive cells (Fig. 2D). These cells co-localize with DAPI (marker for nuclear staining) as well.

Figure 2 Representative images of immunofluorescent analysis and quantification of proliferating cells in the region of the SVZ as revealed by the Ki67 marker in control (con) and TIF-IAD1RCre-mutant (mut) riluzole treated and non-treated mice (A, B). Example of Ki67/MAP2/DAPI (C) or Ki67/DCX/DAPI (D) triple-stainings carried out in attempt to confirm nuclear localization and neuronal origin of Ki67 signal.

Numbers of Ki67+ cells are represented by means ±  S.E.M. (n = 4–6; ∗p < 0.05 vs. con/VEH). RIL, riluzole; VEH, vehicle. Scale bars: 50 µm.

Lack of riluzole efficacy on progression of MSNs cell death despite enhancement of cell proliferation

A comparative analysis of immunohistochemical staining patterns with chosen markers characteristic for neurodegenerative process in MSNs (marker for labeling mature neurons, NeuN; an oxidative stress indicator marker, 8-hydroxydeoxyguanosine, 8OHdG; astrocyte marker, GFAP; marker for degenerating neurons, FluoroJade C) did not show any visual differences between 13-week-old mutant TIF-IAD1RCre mice with or without riluzole treatment (Figs. 3A–3D). The expression of all of the above-mentioned markers seems to be similar in the riluzole treated and non-treated TIF-IAD1RCre-mutant mice, showing comparable enhancement of inflammatory processes, oxidative stress and neuronal loss.

Figure 3 Representative images of immunohistochemical analysis showing staining of striata with the NeuN (A), induction of oxidative stress detected by the anti-8OHdG antibody (B), astrogliosis visualized by the GFAP-specific antibody (C) and degenerating neurons detected by FluoroJade C staining (D) in control (con) and TIF-IAD1RCre-mutant (mut) mice.

RIL, riluzole; VEH, vehicle. Scale bars: 5 µm (A, C, D), 25 µm (B).

Lack of riluzole efficacy on the behavioral phenotype associated with MSN degeneration

Chronic riluzole administration did not prevent impaired motor coordination of 13-week-old mutant TIF-IAD1RCre mice as demonstrated by the rotarod test (Fig. 4). The riluzole administration had no effect on control animals, while the different effect of the introduced mutation is reflected in a 2-way ANOVA, which reveals a treatment (riluzole) ×  genotype interaction for genotype [F(131.38) = 80.39 (p < 0.0001)] but not for riluzole itself.

Figure 4 Assessment of motor coordination of control (con) and TIF-IAD1RCre-mutant (mut) mice demonstrated by endurance in the rotarod test.

Values for endurance time are represented by means ±  S.E.M. (n = 7–8; ∗p < 0.05; ∗∗p < 0.01 vs. con/VEH). RIL, riluzole; VEH, vehicle.

Riluzole does not affect dendritic spine morphology in TIF-IAD1RCre mice

Our previous research done on TIF-IAD1RCre mice clearly showed that although the neurodegeneration (cell loss) is not observed earlier than in 13-week-old animals, some symptoms of cellular impairment can be seen 2-4 weeks in advance (Kreiner et al., 2013). Taking this into account, we checked whether chronic riluzole treatment could have any positive effects on neural cell morphology. Nevertheless, the performed analysis of the morphology of dendritic spines on 9-week-old animals (where no cell loss is observed yet) did not show any differences between mutants and controls (Fig. 5). There was also no effect of riluzole application on dendritic spine morphology in control animals.

Figure 5 Visualization (A–B) and quantification (C) of dendritic spines in the control (con) and TIF-IAD1RCre-mutant (mut) mice.

The spines were counted in the dorsal striatum (Bregma 1.1–0.0). Data are represented by the means ± SEM (n = 3–4). RIL, riluzole; VEH, vehicle.

Discussion

The objective of this research was supported by preliminary studies, in which we observed an increase in cell proliferation within the SVZ in 9-week-old TIF-IAD1RCre mice suggesting the existence of ongoing neurogenesis (Figs. 2A–2B). This assumption is consistent with other studies reporting increased neurogenesis in other models of progressive neurodegeneration (Luzzati et al., 2011; Nato et al., 2015). This prompted us that the progressive TIF-IA-driven neurodegeneration in these mice offers the unique advantage to study if, in such conditions, the endogenous progenitors potentially involved in putative neuroprotective mechanisms can be modulated by experimental treatments. However, when performed double staining with Ki67 and NeuN within the SVZ in 9-week-old TIF-IA D1RCre mice, we were not able to find any evidence of co-localization. On the other hand, further immunofluorescent analysis revealed that the number of cells labelled with Ki67 co-localize with the MAP2 or DCX neuronal markers (Figs. 2C–2D). This may be explained by the fact that DCX and MAP2 belong to early markers of neuronal maturation, while NeuN is a marker of mature neurons being expressed later on (Sarnat, 2013).

To elucidate whether the putative further enhancement of cell proliferation, potentially responsible for evoking adult neurogenesis, induced by chronic riluzole administration can have any positive influence on the progression of the neurodegenerative phenotype observed in TIF-IAD1RCre mice, we evaluated chosen markers of neurodegeneration known to be differentially expressed in these mice at 13 weeks old, where the cell loss starts to be clearly visible as described previously (Kreiner et al., 2013). Additionally, we also screened their behavioral phenotype by assessing motor coordination. Despite an observed increase in newborn cells in the SVZ after riluzole administration as visualized by Ki-67 staining (an effective marker of proliferating cells (Kee et al., 2002)) (Fig. 2), neither experimental approach showed any differences between riluzole-treated and non-treated mutants, revealing a similar extent of the neurodegenerative phenotype evaluated in 13-week-old animals (Figs. 3 and 4). The mice were characterized by the same expression of induced-GFAP and 8OHdG and profoundly reduced staining intensity for NeuN in the striatum (Figs. 3A–3C). The neurodegenerative phenotype was further confirmed by FluoroJade C staining, an effective marker of degenerating neurons (Schmued et al., 2005) (Fig. 3D). This was reflected by the impairment in motor coordination on the rotarod test, and again, no differences were observed between riluzole- and vehicle-treated mutants (Fig. 4). Overall, these experiments did not show any beneficial effects of riluzole administration on the progress of the mutation.

It seemed that further enhancement of this process by riluzole administration can bring considerable benefits in the form of slowing down the progression of the mutation. Our transgenic models based on the conditional ablation of transcription factor TIF-IA have already been positively verified as a possible tool to study the mechanisms of action of other pharmacotherapies. In particular, we showed that the progression of neurodegenerative phenotype in the TIF-IADATCre mice (PD model) can be postponed by L-DOPA (Rieker et al., 2011) or reboxetine treatment (Rafa-Zabłocka et al., 2014).

It can be argued that either the treatment paradigm was not appropriate to achieve the expected drug efficacy or the effects were simply too weak to have any functional meaning. Regarding the first issue, the dose of riluzole in chronic experiments performed on rodents does indeed range from 1 mg/kg to 40 mg/kg (Besheer, Lepoutre & Hodge, 2009; Carbone, Duty & Rattray, 2012; Fumagalli et al., 2006; Sepulveda, Astorga & Contreras, 1999), and is predominantly 20 mg/kg when used to evoke a neurogenesis response. Therefore, the dose used in our experiment was in the lower range of the therapeutic window. The reason for choosing this particular dose was determined by the lethargy and spastic gait followed by a high mortality rate of the mice treated with 20 mg/kg and 10 mg/kg. This problem has also been reported by other researchers when rats were treated with similar doses and exhibited locomotor ataxia and lethargy (Kitzman, 2009; Simard et al., 2012). We presume that this phenomenon is associated with the specific mouse strain (C57Bl/6N) rather than with the introduced mutation since the problem affected both control and TIF-IA D1RCre mice. Nevertheless, it has to be emphasized that even the dose of 5 mg/kg was able to induce cell proliferation within the SVZ region as visualized by Ki67 staining (Fig. 2A) and quantified afterwards (Fig. 2B). Moreover, there are existing reports that prove a similar dose to be effective (Kitzman, 2009).

In addition, in order to evaluate whether riluzole can exert any influence on affected MSNs, we performed a quantitative analysis of dendritic spine morphology at the stage when the neurons were still present in the striata of TIF-IAD1RCre mice. We analyzed 9-week-old mice, as this is the stage where no cell loss has been observed but the cascade of molecular events leading to degeneration has already been prompted (Kreiner et al., 2013). Nevertheless, this analysis did not show any changes in dendritic spine morphology (Fig. 5), supporting the observation that riluzole seems to not be effective in the investigated model.

Surprisingly, we were not able to find any abnormalities in the morphology of dendritic spines in the non-treated TIF-IA D1RCre-mutant mice despite the clear neurodegenerative phenotype that has already been documented. This issue has not been addressed in our previous work. However, the occurrence of such changes is not always correlated with neurodegeneration (Dickstein et al., 2010) or may be a subsequent event. On the other hand, lack of spine pathology might also be attributed to the relatively early stage of pathology observed in 9-week-old TIF-IA D1RCre mutants, as other authors have shown that spine pathology was present in late (36-weeks old) (Spires et al., 2004), but not early (20-weeks old) (Nithianantharajah et al., 2009), symptomatic stages of the R6/1 Huntington disease model.

The lack of analysis of other time points (i.e., 16-week-old animals or older) can be regarded as a drawback of the experimental design. This is mainly due to the relatively low number of animals in the cohort, which is restricted by current strict animal welfare regulations. Nevertheless, since the phenotype of riluzole-treated TIF-IAD1RCre mice is non-distinguishable from untreated mutants (regarding both the behavioral and histological levels) at the age of 13 weeks (where the cells are already starting to degenerate), it would be hard to imagine that any differences would be observed at a later period. Lack of differences at this pivotal stage does not provide any strong support for the investigation of earlier time points, which could have been interesting if we had observed findings differentiating the animals at 13 weeks. However unlikely, it cannot be excluded that analysis of additional time points in between 9th and 13th week would differentiate the riluzole treated and non-treated animals.

In spite of expectations based on previously gathered evidence in preclinical studies and the use of riluzole in clinics for the treatment of ALS, a recent study also yielded disappointing results concerning this drug. Despite being an expensive drug, it does not stop the progression of ALS and is not always well tolerated, making the efficacy of riluzole in the treatment of ALS inconclusive (Ginsberg & Lowe, 2002). Moreover, experiments performed on animal models assessing riluzole as a potential treatment for HD and spinocerebellar ataxia (SCA) had no beneficial effects (Hockly et al., 2006; Schmidt et al., 2016). In clinical trials of anti-HD treatment, there was also no clear neuroprotective effect of riluzole administration, and its effects were narrowed only to reduced chorea (Frank, 2014). Thus, our study seems to be in line with others that question the effectiveness of riluzole in animal models and raise concerns about the utility of this drug due to its rather modest clinical efficacy (Limpert, Mattmann & Cosford, 2013).

Conclusions

Despite an observed increase in newborn cells in the SVZ after riluzole administration, our study did not show any differences between riluzole-treated and non-treated mutants, revealing a similar extent of the neurodegenerative phenotype evaluated in 13-week-old TIF-IAD1RCre animals, a new transgenic model resembling HD-like neurodegeneration. This lack of an observed effect could be due to either the treatment paradigm or the possibility that the effects were simply too weak to have any functional meaning. Nevertheless, this study is in line with others that question the effectiveness of riluzole in animal models and raise concerns about the utility of this drug due to its rather modest clinical efficacy.

Supplemental Information

Supplemental Information 1 Raw data from Ki67+ cells counting

Click here for additional data file.

Supplemental Information 2 Collected data from rotarod motor assessment

Click here for additional data file.

Supplemental Information 3 Collected data from dendritic spines assessment

Click here for additional data file.

We thank Prof. Günther Schütz and Dr. Rosanna Parlato from the German Cancer Research Center (DKFZ, Heidelberg, Germany) for their generous gift of the TIF-IAD1RCre mice.

Additional Information and Declarations

Competing Interests

Author Contributions

Animal Ethics

Data Availability

The authors declare there are no competing interests.

Grzegorz Kreiner conceived and designed the experiments, performed the experiments, analyzed the data, contributed reagents/materials/analysis tools, wrote the paper, prepared figures and/or tables, reviewed drafts of the paper.

Katarzyna Rafa-Zabłocka performed the experiments, analyzed the data, reviewed drafts of the paper.

Piotr Chmielarz performed the experiments, analyzed the data, prepared figures and/or tables, reviewed drafts of the paper.

Monika Bagińska performed the experiments.

Irena Nalepa analyzed the data, reviewed drafts of the paper, supervised the study.

The following information was supplied relating to ethical approvals (i.e., approving body and any reference numbers):

Animal Ethical Committee, Institute of Pharmacology, Polish Academy of Sciences provided full approval for this research (Permit Number: 951, issued: June 28, 2012).

The following information was supplied regarding data availability:

The raw data has been supplied as Supplementary Files.

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
