# Peer review of "Lack of riluzole efficacy in the progression of the neurodegenerative phenotype in a new conditional mouse model of striatal degeneration"

_PeerJ, doi:10.7717/peerj.3240_

## Round 0.1 · original submission · Major Revisions

Both reviewers point out that significant changes to the presentation and interpretation of your results are needed.

Reviewer 1 ·

Basic reporting

1) The authors cannot define the conditional TIF-IA model as a new model of Huntington's disease (HD). They should rather indicate this as: HD-like and be more coherent with this definition in the entire text.
It is true that there is no perfect mouse model of HD and that there are several similarities in the phenotype of the conditional TIF-IA mice and HD, but the TIF-IA based model mimics a condition of nucleolar stress that might be involved in HD as well as in other neurodegenerative disorders.

2) It would be useful to present in more detail the mechanisms through which riluzole works and its current therapeutic use, in particular hypothesis on mechanisms to increase neurogenesis. Different references are mentioned but the text is not explicit enough. On the other hand the authors write: “Nevertheless, there are still concerns about its utility due to its rather modest clinical efficacy”. A reference should be provided here.

3) The increased neurogenesis observed in the TIF-IA model is presented as based on preliminary results (see introduction and results), while this is a novel finding here. The authors could put their work in the context of papers reporting increased neurogenesis in other models of progressive neurodegeneration such as:
- Nato G, Caramello A, Trova S, Avataneo V, Rolando C, Taylor V, Buffo A, Peretto P, Luzzati F. Development. 2015 Mar 1;142(5):840-5. doi: 10.1242/dev.116657.
- Luzzati F, De Marchis S, Parlato R, Gribaudo S, Schütz G, Fasolo A, Peretto P. PLoS One. 2011;6(9):e25088. doi: 10.1371/journal.pone.0025088.
Actually they write: “Thus, the progressive TIF-IA-driven neurodegeneration in these mice offers the unique advantage to study if, in such conditions, the endogenous progenitors potentially involved in putative neuroprotective mechanisms can be modulated by experimental treatments.”, but somehow they are not coherent with this concept in the manuscript.

Experimental design

Results: this part should be significantly revised, in fact I suggest to show first the increased neurogenesis in the different experimental groups by Ki67 and including a quantification to support the lack of changes between the groups. Increased neurogenesis upon striatal neurodegeneration in the TIF-IA mutant is per se a positive and novel finding. Also the authors should clearly indicate what phenotype corresponds to which stage and also at which stages each experiment was performed. This would help to better understand the study design and rationale behind this work. In the current form the text is too concise. Also the results on the TIF-IA models lacking spine defects should be highlighted in the context of previous work with these mice.
Minor point: NeuN is marker for mature neurons and not immature.

Validity of the findings

Discussion: I disagree on the “Moreover, lack of differences at this pivotal stage does not provide any support for the investigation of earlier time points, which could have been interesting if we had observed findings differentiating the animals at 13 weeks.” This sentence should be modified to discuss the possibility of using earlier time points (cell death occurs at 13 weeks, it might be too late to intervene!). Also on this point, more detailed information on the TIF-IA mice and time course of phenotypic alterations is missing and would help to discuss the results.

Additional comments

In this manuscript Kreiner et al. investigate the impact of riluzole treatment in a mouse model of progressive striatal neurodegeneration at the behavioral and neuropathological level. This model is characterized by impaired synthesis of ribosomal RNA in striatal neurons and mimics some features of Huntington's disease. Here the authors show increased neurogenesis in this model, however riluzole does not ameliorate rotarod performance and does not reveal a neuroprotective role under the tested conditions.
Despite the negative results, the study represents a step further towards the characterization of a recently developed and potentially useful model of progressive striatal neurodegeneration and adds novel information on the therapeutic limits of riluzole under certain settings.
However, there are several points that need to be addressed before acceptance for publication. These issues are linked to the need to better clarify conceptual aspects and rationale of models and analyses, as indicated in detail

Reviewer 2 ·

Basic reporting

-The language seems OK.
-Introduction and background regarding the model are correct in general, but troubles appear with concepts related to markers employed and with their correct election in view of the questions formulated. For instance,
1- Authors claim that their model causes the death of medium spiny neurons but they used only a pan-neuronal marker that recognizes all of the striatal neuronal population;
2- It is employed GFAP, a classical astrocyte marker and other indicator of oxidative stress, as directly correlated with neurodegeneration but did not employ any direct marker of brain damage. Authors neither evaluate neuronal density in dorsal striatum to show neuronal death;
3- Ki67 is consensually accepted as a marker of cell proliferation that in fact one of the most useful indicators employed for tumor cell proliferation. However, in the text it is mentioned several times as a marker of neurogenesis when this is right in the sole condition in which Ki67 is used together with a neuronal marker.
-Literature is well referenced and relevant, probably a bit long considering the extension of the manuscript.
-Structure conforms to PeerJ standard, discipline norm, or improved for clarity: it seems fine.
-Regarding figures, definitely, they are not relevant and not informative because did not show the claims purposed. My judgement is based on:
- Figure 1A shows the labeling of NeuN suggesting widespread decrease in the number of striatal neurons; therefore other neurons besides medium spiny neurons seem affected. Authors must analyze this result;
-in Figure 1B the third image is out of focus;
-in Figure 1C, the signal of GFAP is dramatically different in the first two images when compared with the other two. In fact in the first two, staining evidenced myelinated striatal bundles. These bundles are typical of the striatal histo-architecture completely disappeared from images 3 and 4 of the same panel. Is myelin also affected? The authors absolutely ignored all of these results in their consideration;
-typical Ki67 signal is nuclear. In most of Figure 1D, it is very hard to recognize any nuclei and to differentiate real nuclear signal from the background or fragmented positive spots;

Experimental design

-Regarding originality of the research and fitting to the journal scope: it seems fine.
-Research question is well defined and relevant. However, in my opinion, the strategy employed to answer the formulated questions is wrong.
-I cannot give an opinion about the rigorous investigation performed; however the results did not reflect it.
-Methods were not described with sufficient detail. Experiments cannot be replicated with information wrote in the manuscript.

Validity of the findings

-Impact and novelty: the paper is of potential interest but results showed did not reflect it. -Negative/inconclusive results accepted: in my opinion this happens in the whole manuscript.
-Meaningful replication encouraged where rationale & benefit to literature is clearly stated: Not satisfactory.
-Data is not robust and statistically incomplete.

Additional comments

The aim of the manuscript is interesting and sound as well as the experimental paradigm employed. However the questions and the approaches chosen to answer these questions are inappropriate. There are also a number of wrong concepts that are employed in the whole text.

Major points:
-Under the title Methods in the abstract GFAP and 8OHdG are named as “specific markers associated with neurodegeneration”. In my opinion this is wrong. In spite that the immunoreactivity of GFAP, the most classical marker of astrocytes, usually increases in several neurodegenerative conditions, the solely increased expression of GFAP is not enough to assume that neurodegeneration occurs. In the same line, 8OHdG is a marker of oxidative stress but insufficient to evidence neurodegeneration. Authors must employ other tools to claim the neurodegenerative stage, i.e. by determining the density of striatal NeuN positive cells in all experimental conditions, employ markers of apoptosis such as cleaved caspase 3 or markers of neurodegeneration such as FluoroJade B or C to give validity to their findings.
-The phrase “A comparative analysis of IHC staining patterns with chosen markers for the neurodegeneration process...” that appears in Results in the abstract must be corrected. Firstly, the authors only employed an astrocyte marker and an oxidative stress marker, both seeming insufficient to assume that neurodegeneration occurs. They must at least complete the data with by quantifying neuronal density that seemed decreased in the images shown.
Secondly, experiments were performed at a single time then it cannot be assumed that neurodegeneration process was followed.
-In Results from the Abstract appears that Ki67 is a marker of neurogenesis. This must be corrected and written properly. Ki67 is a marker of cell proliferation. To determine if neurogenesis increased, a double immunostaining by using Ki67 together with a neuronal marker must necessarily be performed.
-In my opinion, the authors cannot assume there are neurodegeneration with the astrocyte and oxidative stress markers employed.
-Line 172 and Figure 1C: The labelling of GFAP is clearly different in the images corresponding to control/vehicle and control/riluzole as compared to mutant/vehicle and mutant/riluzole. From my experience in the first two images, this is not the right GFAP signal, because axonal myelinated bundles appear positive. Because such bundles must be negative to GFAP, this indicates a complete lack of specificity of the signal. GFAP signal in the third and fourth image of Fig. 1C seemed more in accordance with a reactive gliosis but axonal bundles that must be negative to the signal and are typical in the striatum do not appear in both images.
-Line 185: In my opinion the authors cannot claim that the dendritic morphology of mutant mice is not affected because they did not show any image of this finding. Moreover, absence of morphological neuronal effects seem unlikely in the case of massive death as suggested by decreased striatal NeuN staining found. If the authors would like to claim that morphology of surviving neurons looked normal, they must show it and this work will be enriched very much. This claim cannot also be sustained in discussion.
-Line 204: It cannot be said that GFAP expression did not change when images did not support this sentence.
-Fig. 1D: Regarding the labelling of Ki67, magnification of the images did not allow seeing positive nuclei. Therefore, images must be shown at higher augments or the Ki67 signal must be shown together with a nuclear marker (such as DAPI) to distinguish positive nuclei from background non sense signal. At this respect, in the images that appear more positive signal, it also appears increased background.

Minor points
-In Methods in the Abstract appears a phrase that says neuronal markers were used. The phrase must be fitted since only one neuronal marker (NeuN) was employed.
-Line 128: DMSO concentration employed seems very high mostly in view of 14 consecutive injections.
-Line 139-140: It is not clearly stated when animals were processed. The sacrifice was immediately after the last treatment or 24 h later?
-Line 141: The thickness of sections employed to immunolabeling must be clearly stated. -Line 171: NeuN is generally considered as a marker of mature postmitotic cells.
-Line 175: Ki67 cannot be considered as a marker of neurogenesis, it is a marker of cell proliferation.
-Line 156: In view of the size of the striatum, quantitating dendrites of 5 neurons per animal seems insufficient to obtain reliable data.

---

## Round 0.2 · Minor Revisions

Please include extra information regarding the neurodegeneration mentioned in your rebuttal letter (e.g. by sumarizing the characterization you described earlier in Kreiner et al. 2013) . You should also further support your claim of co-localization of MAP2 and Ki67.

Reviewer 2 ·

Basic reporting

Language and grammar: OK
References: Very good
Figures and tables: Have improved respect to the initial version

Experimental design

-The improvements made in the revised version enriched the work.
-Details of procedures are now clear and friendly for the reader.

Validity of the findings

Results added are very interesting.
In my opinion, the claim of increased neurogenesis is not supported by the experimental data. This needs to be shown indubitably and is not the case likely because of the atypical immunoreactivity against MAP2 (Fig. 2C).

Additional comments

GENERAL COMMENTS
The aim of the manuscript is interesting and sound as well as the experimental paradigm employed. However the questions and the approaches chosen to answer these questions are inappropriate and remain inappropriate in the revised version in spite that authors made some improvements to the text and add relevant details to Material and Method section.

Major points that needs to be corrected:
-Authors insist to claim that “GFAP and 8OHdG are “specific markers associated with neurodegeneration”. They completely ignore the evidence that sustain that the immunoreactivity against GFAP, a classic marker of astrocytes, usually increases in several neurodegenerative conditions, but decreases in others1-3.
-Authors did not directly evaluate neurodegeneration neither by estimating the density of living neurons as suggested previously or adding a direct evidence of neuronal death that can be easily done by using Fluorojade B or C, Annexin V or more generally by a Tunnel assay.
-In Fig. 2C, authors say that MAP2 co-localizes with Ki67. In my opinion, co-localization is not supported because that MAP2 usually recognizes neuronal processes4 that cannot be recognized in the image even at higher magnifications. Co-localization is crucial to claim the existence of increased neurogenesis. Therefore, the lack of indubitably evidence of co-localization between Ki67 and a neuronal marker destroys the claim of increased neurogenesis. On the other hand, if it is intended to show increased neurogenesis probably the best marker to use with Ki67 is TujIII5.

REFERENCES
1 Dietrich J, Lacagnina M, Gass D, Richfield E, Mayer-Pröschel M, Noble M, Torres C, Pröschel C (2005). EIF2B5 mutations compromise GFAP+ astrocyte generation in vanishing white matter leukodystrophy. Nat Med. 2005 Mar;11(3):277-83. Epub 2005 Feb 20.
2 Hol EM, Pekny M (2015) GFAP Glial fibrillary acidic protein (GFAP) and the astrocyte intermediate filament system in diseases of the central nervous system. Curr Opin Cell Biol. 32:121-130.
3 Hostenbach S, Cambron M, D'haeseleer M, Kooijman R, De Keyser J (2014) Astrocyte loss and astrogliosis in neuroinflammatory disorders. Neurosci Lett. 17;565: 39-41.
4 Vessoni AT, Herai RH2, Karpiak JV3, Leal AM4, Trujillo CA3, Quinet A5, Agnez Lima LF6, Menck CF5, Muotri AR7 (2016) Cockayne syndrome-derived neurons display reduced synapse density and altered neural network synchrony. Hum Mol Genet. 25(7):1271-1280.
5 Long K, Moss L1, Laursen L2, Boulter L3, ffrench-Constant C1 (2016) Integrin signalling regulates the expansion of neuroepithelial progenitors and neurogenesis via Wnt7a and Decorin. Nat Commun 7:10354.

---

## Round 0.3 · accepted · Accept

I am satisfied with your revisions.